# The Endothelial Glycocalyx as a Target of Ischemia and Reperfusion Injury in Kidney Transplantation—Where Have We Gone So Far?

**DOI:** 10.3390/ijms22042157

**Published:** 2021-02-22

**Authors:** Anila Duni, Vassilios Liakopoulos, Vasileios Koutlas, Charalampos Pappas, Michalis Mitsis, Evangelia Dounousi

**Affiliations:** 1Department of Nephrology, School of Health Sciences, University of Ioannina and University Hospital of Ioannina, 45110 Ioannina, Greece; anikristduni@yahoo.com (A.D.); harpapdr@yahoo.gr (C.P.); 2Division of Nephrology and Hypertension, 1st Department of Internal Medicine, Medical School Aristotle University of Thessaloniki, 54636 Thessaloniki, Greece; liakopul@otenet.gr; 3Kidney Transplant Unit, Department of Surgery, University Hospital of Ioannina, 45500 Ioannina, Greece; v_koutlas@yahoo.com (V.K.); mitsism@hotmail.com (M.M.)

**Keywords:** kidney transplantation, endothelial glycocalyx, ischemia and/or reperfusion injury, inflammation, immune responses

## Abstract

The damage of the endothelial glycocalyx as a consequence of ischemia and/or reperfusion injury (IRI) following kidney transplantation has come at the spotlight of research due to potential associations with delayed graft function, acute rejection as well as long-term allograft dysfunction. The disintegration of the endothelial glycocalyx induced by IRI is the crucial event which exposes the denuded endothelial cells to further inflammatory and oxidative damage. The aim of our review is to present the currently available data regarding complex links between shedding of the glycocalyx components, like syndecan-1, hyaluronan, heparan sulphate, and CD44 with the activation of intricate immune system responses, including toll-like receptors, cytokines and pro-inflammatory transcription factors. Evidence on modes of protection of the endothelial glycocalyx and subsequently maintenance of endothelial permeability as well as novel nephroprotective molecules such as sphingosine-1 phosphate (S1P), are also depicted. Although advances in technology are making the visualization and the analysis of the endothelial glycocalyx possible, currently available evidence is mostly experimental. Ongoing progress in understanding the complex impact of IRI on the endothelial glycocalyx, opens up a new era of research in the field of organ transplantation and clinical studies are of utmost importance for the future.

## 1. Introduction

Kidney transplantation, the treatment of choice for end-stage kidney disease, is associated with remarkable improvements in patients’ prognosis, including survival, cardiovascular outcomes and quality of life, as compared to dialysis [1,2]. Despite tenable improving trends relating to long term kidney allograft survival, the rates of chronic graft loss following the first year after transplantation remain considerable [3,4]. Apart from immunological culprits including human leukocyte antigen (HLA) mismatching and sensitization, the type of kidney donor, long-term immunosuppression, as well as comorbidities such as arterial hypertension and dyslipidemia, large studies suggest that perioperative factors are implicated in the augmented risk of long-term allograft failure [5,6,7,8,9,10,11]. Ischemia and/or reperfusion injury (IRI) in kidney transplantation is at the forefront as a critical risk factor associated not only with early complications such as delayed graft function in the setting of post-ischemic acute tubular necrosis but with acute rejection and long-term allograft dysfunction as well [12]. IRI, at least at some degree, is an inevitable phenomenon occurring during kidney transplantation. Although the term at its core denotes a disturbance of blood flow, nevertheless there is a multitude of intertwining pathophysiological pathways underlying the complex pathological and clinical implications of this entity [13,14,15].

There is abundant information available in the literature with regard to kidney IRI, including numerous experimental and clinical studies which attempt to shed light on the intricate mechanisms involved in its pathogenesis as well as its main targets, the vascular endothelium and renal tubular epithelial cells. Among others, the negatively-charged carbohydrate-rich gel-like structure known as the endothelial glycocalyx, which lies at the interface between blood and the endothelium has come at the spotlight of extensive research, due to its fundamental role in the maintenance of endothelial homeostasis. Glycocalyx should not be considered as a mere mixture of proteoglycans, glycoproteins, and glycolipids. It plays a pivotal modulatory role in endothelial function, not only due to its biomechanical properties which regulate shear stress transduction to the endothelium, but due to its composition as well, which includes proteins involved in cell attachment and migration, growth factors, chemokines, mediators of oxidative stress and coagulation factors [16].

## 2. Aims and Methods

The aim of our review is to present the currently available data regarding complex links between shedding of the glycocalyx components, like syndecan-1, hyaluronan, heparan sulphate, and CD44 with the activation of intricate immune system responses, including toll-like receptors, cytokines and pro-inflammatory transcription factors. Evidence on modes of protection of the endothelial glycocalyx and subsequently maintenance of endothelial permeability as well as novel nephroprotective molecules such as sphingosine-1 phosphate (S1P), are also depicted. Accordingly, we searched the electronic databases including PubMed, Medline, and Cochrane for all publications on solid organ transplantation or kidney/renal transplantation, and ischemia and reperfusion injury and acute kidney injury and endothelial glycocalyx, syndecan, hyaluronan, heparan sulphate, CD44, until November 2020. We included both experimental and original clinical studies. Additionally, we hand-searched the references of every relevant study and review article for additional publication.

## 3. IRI at a Glance

Ischemia and reperfusion injury represent an invariable and major challenge during the perioperative period in kidney transplantation. The time path of the events which determine the extent of IRI in this setting, from brain death and associated sympathetic nervous system hyperactivity, to warm ischemia after clamping of the kidney vessels and cold ischemia after graft refrigeration until graft implantation and reperfusion, has a common denominator which is defined by reduced oxygen and nutrients supply to the renal tissue [13]. The ensuing switch to anaerobic glycolysis, fails to meet the energetic demands of the renal cells, leading to leakage of lysosome enzymes due to disruption of the lysosomal membrane, inhibition of the Na^+^/K^+^/ATPase activity and calcium overload within the cytoplasm [14,17,18,19]. Paradoxically, the process of reperfusion itself in the setting of this ischemic milieu ignites the generation of reactive oxygen species (ROS) and the activation of intracellular calcium-dependent proteolytic enzymes, thus perpetrating further damage [20,21]. The simplistic approach depicted above is a universal process common to all cells exposed to an ischemic environment; however it entails the participation and integration of several distinct cellular and molecular pathways, including cell-death programs such autophagy, necroptosis and apoptosis, the activation of the pro-inflammatory cascade, endothelial dysfunction manifesting as augmented expression of vasoactive and vascular adhesion molecules and amplification of the oxidative stress [22,23,24,25,26,27,28].

The innate immune system and in specific activation of toll-like receptor (TLR)—4 on white blood cells as well as on endothelial and renal tubular cells plays a key role in IRI, leading to a cascade of increased expression of pro-inflammatory transcription factors, NF-kB and activator protein 1. Upregulation of adhesion molecules, including intracellular cell adhesion molecule (ICAM-1), vascular cell adhesion molecule VCAM-1, and E-selectin, which facilitate leukocyte migration and infiltration, further augment the inflammatory response and the activation of the immune system [15,29,30,31]. Activation of TLR-4 has been shown to promote the release of major pro-inflammatory cytokines such as interleukin (IL)-6, IL-1, tumor necrosis factor (TNF) and chemotactic mediators such as macrophage inflammatory protein-2 (MIP-2) and monocyte chemo attractant protein-1 (MCP-1) [32]. Additionally, there is an interaction between TRL signaling and the complement system in IRI with mitogen activated protein kinases (MAPKs) serving as the linking chains between the two systems [15,33]. Furthermore, following recognition of cellular debris released in the setting of cellular injury, the so-called danger-associated molecular patterns (DAMPs), TLRs activate not only the inflammatory response as depicted above, but induce the dendritic cells to perform their antigen-presenting role to the B- and T-lymphocytes of the adaptive immune system [34]. In the setting of IRI, renal TLR-4 recognizes among other endogenous ligands, molecules of the extracellular matrix and the glycocalyx such as biglycan, hyaluronan and heparan sulfate [35,36,37].

Reactive oxygen species are pivotal components of the pathogenesis of IRI. Ischemic deregulation of mitochondrial function most probably causes an outbreak of ROS release due to activation of xanthine oxidase and nicotinamideadenine dinucleotide phosphate (NADPH) oxidase following reperfusion and reinstallation of proper tissue oxygenation. An imbalance between ROS and reactive nitrogen species (RNS) formation and the endogenous antioxidant system responses ensues, resulting in oxidative damage and further activation of inflammation as well as pro-apoptotic mediator expression, thus perpetuating a vicious circle [28,38,39,40,41]. Intracellular metabolic byproducts accumulated during the ischemia phase, such as succinate, have been shown to further disrupt mitochondrial electron transport and induce superoxide generation [42].

Hypoxia inducible factors (HIF), HIF-1 and HIF-2, are transcription factors which have gained a lot of attention due to their potential beneficial role in IRI [15,43]. Von Hippel–Lindau and prolyl hydroxylase domain (PHD) proteins are associated with HIF degradation during conditions of normoxia [43]. On the other hand, HIF are stabilized by hypoxia, thus regulating tissue adaptation to such conditions through transcription of target genes, including those associated with glycolysis, production of angiogenic factors such as the VEGF and erythropoietin production [43]. It should be noted that HIF-1 upregulate TLR4 expression in macrophages in response to hypoxic stress whereas ROS mediate HIF-1 regulation through distinct pathways, including inhibition of propyl hydroxylases, post-translational modification of HIF-1a protein by the process of nitrosation, as well as indirectly through involvement of miR-21, miR-210 and inflammatory mediators [44,45].

A major target of IRI and the associated pathogenic processes is vascular endothelial dysfunction, manifesting as endothelial cell swelling, degradation of the endothelial cytoskeleton, loss of endothelial layer integrity as well as glycocalyx degradation which shall be elaborated subsequently in detail [46,47]. The culminating event is the endothelial-to-mesenchymal transition (EndMT), during which the endothelial cells display a phenotype similar to mesenchymal cells, demonstrated by an increased propensity for increased extracellular matrix production and migratory properties [48,49].

## 4. An Overview of the Endothelial Glycocalyx

The backbone of the endothelial glycocalyx consists of proteoglycans together with their polysaccharide chains of glycosaminoglycans (GAGs) as well as glycoproteins and glycolipids [16,50]. The main GAG constituents are heparan sulfate (HS) and chondroitin sulfate (HS) which are attached to proteoglycans whereas hyaluronic acid (HA) directly binds to CD44, a transmembrane glycoprotein. The family of syndecans (including syndecan-1, syndecan-2, syndecan-3 and syndecan-4) represents single transmembrane domain proteoglycans whereas glypican-1 is an extracellular glycosylphosphatidylinositol (GPI)-anchored HS glycoprotein [51]. Additionally, perlecan and biglycan are soluble form of proteoglycans which reside within the glycocalyx matrix without being attached to the endothelial cell membrane. It has been shown that the glycocalyx thickness and composition differ among various organs, vascular anatomic sites and even within fenestrated versus non-fenestrated capillary beds, which in turn might determine heterogeneous glycocalyx properties respectively [16,52]. Vascular shear stress and sphingosine-1-phosphate (S1P), a phospholipid that participates in signaling pathways mediated by G-protein–coupled receptors appear to be significant determinants and regulators of the glycocalyx structure and function [53].

Furthermore, glycoproteins are also considered as essential functional constituents of the glycocalyx, because they add to its diverse biological functions [51]. The main glycoprotein classes include the endothelial cell adhesion molecules and components of the coagulation and fibrinolysis system [51]. Accordingly, E-selectin and P-selectin mediate the interaction between white blood cells and endothelial cells whereas integrins mediate endothelium interaction with components of the extracellular matrix [54,55]. ICAM-1 and -2 as well as VCAM-1 belong to the immunoglobulin superfamily of transmembrane glycoproteins which serve as ligands for integrins on white blood cells and platelets, thus participating in leukocyte trafficking, including recruitment and extravasation to inflamed sites [51,56]. The von Willebrand factor receptor or otherwise glycoprotein Ib-IX–V complex and thrombomodulin, a thrombin co-factor and natural anti-coagulant represent among others, membrane-bound proteins of the endothelial glycocalyx with a regulatory role in coagulation and fibrinolysis.

Apart from cell-membrane anchored proteins, there is an abundance of numerous molecules from various origins (e.g., plasma, endothelial cells, etc.) residing within the glycocalyx microenvironment. These soluble components include enzymes belonging to the defense system of the organism against ROS (superoxide dismutase), interleukins, fibroblast growth factor (FGF) and transforming growth factor b (TGFb), LDL lipase, and members of the coagulation cascade such as antithrombin III, and tissue pathway factor inhibitor [51].

Acknowledging the complexity of the endothelial glycocalyx, renders it straightforward to comprehend its pleiotropic properties as a transducer of vascular shear stress forces into the endothelial cells, regulator of vascular permeability, modulator of inflammatory responses and oxidative stress, as well as regulator of hemostasis [51].

Experimental data in the literature suggest that the endothelial glycocalyx is a significant component of the glomerular filtration barrier [16]. Thus, not only does its mesh of negatively charged GAG and sialic acid residues of glycoproteins provide both a charge and size selective barrier, it has been shown that the glomerular endothelial fenestrae are filled with HA, which prevents albumin form transversing the glomerular capillary wall [52,57]. Endothelial deletion of hyaluronan synthase 2 (Has2) in mice is associated with mesangiolysis, glomerular capillary rarefaction, glomerulosclerosis and albuminuria, findings with direct implications in several disease models, including diabetic nephropathy [57].

Likewise, mice deficient in N-deacetylase-N-sulfotransferase (Ndst) enzyme, which modulates HS structure, appear to be protected from glomerular leukocyte influx, in an experimental model of anti-glomerular basement membrane nephritis [58].

Furthermore, it has been shown that HA, HS and glypican-1 are required for the vasoactive response of the endothelial cells to shear stress and in specific through transferring forces to the actin cytoskeleton as well as through endothelial nitric oxide synthase (eNOS) activation and nitric oxide (NO) generation [53,59,60].

It should be noted that the process of studying the endothelial glycocalyx ex vivo and in vitro is a challenging task due to its fragility of structure as well as technical difficulties in preparation of results. Circulating markers of the endothelial glycocalyx might serve as attractive alternatives, as occurs with pathological glycocalyx shedding in various disease models [16].

## 5. IRI and Glycocalyx Damage in Kidney Transplantation

The damage of the endothelial glycocalyx and the related endothelial dysfunction as a consequence of IRI, is common to several disease models. Accordingly, both generalized ischemia states as occur with cardiac arrest and various types of shock or local organ ischemia like in the setting of myocardial infarction and revascularization procedures, are characterized by direct or indirect evidence of endothelial glycocalyx degradation [61]. Likewise, ischemic acute kidney injury (AKI) and IRI in kidney transplantation which manifests as delayed graft function, share common pathophysiological traits, with damage to the glycocalyx being one of those (Figure 1).

### 5.1. IRI Induced Shedding of the Renal Endothelial Glycocalyx

The disintegration of glycocalyx induced by IRI is the crucial event which exposes the denuded endothelial cells to further inflammatory and oxidative damage. Evidence suggests that glycocalyx shedding is a common occurrence and correlates with graft injury in liver and lung transplantation [62,63,64,65]. Thus, plasma syndecan-1 level increases significantly following reperfusion during orthotopic liver transplantation and moreover it predicts superimposition of post-transplant AKI stage 2 or 3 within 48 h following reperfusion [62]. According to recent data, the glycocalyx damage is installed within human liver grafts as early as during graft preservation as indicated by elevated syndecan-1 levels in effluents of liver grafts, which further correlated with effluent concentrations of hepatic injury markers as well as with increased risk for development of early allograft dysfunction [63].

Likewise, increased concentrations of endothelial glycocalyx breakdown products, like syndecan-1, hyaluronan, heparan sulphate, and CD44 have been detected in the perfusate of human and porcine lungs undergoing ex vivo lung perfusion, a new technique which aims to improve graft function in lung transplantation [64]. Decreased hyaluronan levels in the peripheral blood of lung donors were shown to be independently associated with the odds of lungs being acceptable for transplant whereas high levels of plasma syndecan-1 in both lung donors and recipients have been associated with primary graft dysfunction [65]. A lung autotransplantation experiment in pigs, revealed decreased lung tissue levels of syndecan-1 and heparan sulphate together with increased levels in the plasma samples following pulmonary artery clumping and subsequently post-reperfusion, which were additionally accompanied by neutrophil activation and augmented expression of adhesion molecules [66].

Activation of the complement cascade has been directly implicated in the renal tissue damage in the setting of IRI causing endothelial activation with increased VCAM-1 expression and recruitment of inflammatory cells [67,68,69]. In a model of IRI induced in single kidney mice, pharmacological C5 blockade resulted in reduced renal endothelial glycocalyx shedding as manifested by preserved expression of renal vascular HS and reduced circulating syndecan-1 and hyaluronan levels [69].

In kidney transplantation, measurement of syndecan-1 and heparan sulfate concentrations at 5 min following reperfusion of kidneys from DCDs demonstrated increased levels in the transplant renal vein in comparison to the systemic arterial circulation [70]. Remarkably, graft function on the first day after transplantation was inversely associated with the renal flux of syndecan-1 at 5 min after reperfusion.

Glycocalyx components are cleaved from the endothelial cell surface by a variety of matrix proteinases, known as the “sheddases”. Matrix metalloproteinases (MMPs) are a big family of proteolytic endopeptidases, which degrade collagens and other extracellular matrix proteins, thus exerting a multitude of essential physiological functions, from wound healing to angiogenesis [71]. There is a large body of evidence implicating MMP in acute kidney injury and fibrotic kidney disease models in both native and transplanted kidneys [72,73,74,75,76]. Different MMPs have been identified as early biomarkers of severe AKI whereas on the other hand they are suggested to promote renal tubular regeneration following AKI [73]. In an experimental model of IRI associated AKI induced in mice, MMP-2 and MMP-9 activities as well as severity of AKI augmented with increasing ischemia duration. Additionally, MMP-2 expression in the peritubular capillaries of the outer medulla, correlated with outer medulla apoptosis and necrosis [73]. Similarly, examination of the perfusate from human perfused kidneys has shown significantly higher levels of MMP-2 and MMP-9 in perfusates from Donation after Circulatory Determination of Death (DCDD) donors compared to donors after brain death (DBD) [77]. Furthermore, it should be noted that the levels of MMP-2 and MMP-9 were approximately double in DGF kidneys compared to the non-DGF kidneys [77]. Comparable results from lung perfusates during ex vivo human lung perfusions have shown a strong positive correlation between MMP-2 activity, and increased syndecan-1 and hyaluronan concentrates [65].

Increased urine MMP-9 concentration in the first post-operative day following kidney transplantation has been shown to correlate not only with tubular atrophy and fibrosis in renal biopsies performed 3 and 12 months after transplantation but with early as well as long-term graft dysfunction [78]. Additionally, patients with DGF, which is a direct clinical outcome of IRI, display higher urine levels of tissue inhibitors of matrix metalloproteinase (TIMP)-1 and TIMP-2 [78].

Apart from the degradation of the extracellular matrix, the endothelial glycocalyx and the type IV collagen basement membrane of the glomeruli, MMP-9 also seems to possess direct pro-inflammatory properties through activation of IL-8 and the endothelial origin epithelial cell derived neutrophil activating peptide (ENA) −78 [79].

It should be noted that MMP-mediated shedding of syndecans has been found to contribute to the endothelial glycocalyx disruption as occurs in various entities, including diabetes mellitus and other pro-inflammatory states [80,81]. Additionally, bilateral renal IRI in syndecan-1–deficient mice as compared to wild-type mice has been associated with augmented macrophage and myofibroblast numbers as well as tubular injury [82]. Furthermore, several experimental data support that MMP-7 shedding of syndecan-1/CXCL1 complexes from various cell surfaces stimulate neutrophil activation and migration in various tissues [83].

GAG chains of syndecan-1 act as binding sites for hepatocyte growth factor (HGF), and mediate HGF interaction with its specific receptor, the mesenchymal epithelial transition factor (c-Met). In turn, HGF receptor together with its downstream effectors, AKT and glycogen synthase kinase-3β (GSK-3β) have been shown to play a renoprotective role in AKI [81,84]. Pharmacological inhibition of syndecan-1 shedding in the setting of IRI activates phosphorylation of the c-Met/AKT/GSK-3β signaling pathway, thus further supporting the important role of syndecan-1 as a coreceptor for HGF to attenuate apoptosis and inflammation in IRI [85]. Thus, administration of GM6001, a sheddase inhibitor in mice with IRI induced AKI, attenuated the stimulatory effect of IRI on IL-6 and TNFα mRNA levels as well as inhibited syndecan-1 shedding and apoptosis of proximal tubular cells [85].

Considering that reactive oxygen species hold a crucial and common position in the pathogenesis of several models of kidney disease, it would be straightforward to acknowledge their role in the rarefaction and degradation of the microvascular endothelial glycocalyx induced by IRI [86,87,88]. Thus, available experimental evidence suggests that ROS do not affect the biosynthesis of the glycocalyx components, but rather they directly cause shedding of heparan sulphate containing glycosaminoglycans. Accordingly, exposure of conditionally immortalized human endothelial cells to hydrogen peroxide was associated with increased levels of radiolabeled glycosaminoglycans fractions in the cell supernatant, as shown by liquid chromatography and immunofluorescence techniques [87]. Similarly, amplification of oxidative stress has been associated with stimulation the expression and activity of MMP-2 and MMP-9, downregulation of TIMP-1 and TIMP-3 and shedding of the extracellular domain of syndecan-1 from the endothelial cell surface [88].

Syndecan molecules and especially syndecan-1 have been extensively studied in carcinogenesis for their proangiogenic properties mediated by modulation of VEGF-VEGFR-2 signaling [89]. Immunofluorescence staining and co-immunoprecipitation analysis of glomerular cultures showed that syndecan-1 co-localizes and interacts with VEGFR receptor (VEGFR)-2 in endothelial cells in vivo and in vitro, thus in fact serving as e VEGFR coreceptor [90]. Notably, western blotting analysis of animal models with hypoxia induced ischemic AKI showed reduced syndecan-1 expression in the glomerular endothelial cells, which was associated with activation of caspase-3, mediated endothelial cell apoptosis. Down-regulation of syndecan-1 in the ischemic glomeruli prevented clathrin mediated VEGF-dependent endocytosis of VEGFR-2 and as a consequence VEGF signaling, thus leading to endothelial cell dysfunction and apoptosis [90]. VEGF signaling which is essential for the protection of the microvascular structure of the kidneys is down-regulated in the setting of renal IRI [91,92]. Recent evidence from a longitudinal study of kidney transplant recipients as well as animal models showed that increased levels of soluble fms-like tyrosine kinase 1 (sFlt-1), a natural circulating antagonist of VEGF corelate with diminished peritubular capillary area following IRI as well as with higher risk of delayed graft function and graft rejection, impaired graft function, and death [93].

Considering, syndecan-1 property of binding of growth factors and cytokines, it would be straightforward to comprehend current evidence linking increased epithelial syndecan-1 in renal allografts with lower interstitial inflammation, proteinuria and serum creatinine levels as well as improved allograft survival [83].

Yet, it should be noted that the available evidence regarding involvement of the glycocalyx components in the pathogenesis of IRI of kidney transplantation remains controversial and in general not direct. Thus, recent data from renal protocol biopsies as well as from an experimental model of kidney transplantation in rats injected with monoclonal rat anti-mouse syndecan-1 antibodies indicated very low expression of syndecan-1 in the vascular endothelium. Accordingly, the authors suggest that increased levels of plasma syndecan-1 following graft injury should be ascribed to upregulation of tubular syndecan-1 and its partial cleavage by sheddases, such as ADAM17 and MMP-9 [94]. On the other hand, Lu et al., detected syndecan-1 expression mainly in the renal corticomedullary junction, which is the most vulnerable zone to IRI injury as well as both on the basolateral and the luminal side of renal tubular cells, via immunohistochemical study of kidneys from sham-operated and IRI mice. Yet, the authors point out that despite the absence of direct evidence linking syndecan-1 with the renal endothelial structure in their study, future investigation is deemed necessary considering the technical difficulties we currently face for appropriate glycocalyx study as well as the important protective role of the endothelial glycocalyx layer in IRI [85]. Acknowledging that circulating erythrocytes may transiently penetrate the endothelial glycocalyx, which is reflected as a dynamic range of the erythrocyte column width may permit us to indirectly estimate the glycocalyx dimensions. Accordingly, microscan sidestream darkfield imaging of the cortical peritubular microcirculation of human kidney grafts, revealed a reduced dynamic range of the erythrocyte column width at 5 min following reperfusion in kidneys from DCD as compared to living donor kidneys. It would be straightforward for us to interpret this fact as a significant loss of the glycocalyx layer early in the course of renal ischemia and reperfusion after kidney transplantation [70].

### 5.2. A Closer Inspection of Heparan Sulfate and Hyaluronan

The HS moieties of the endothelial glycocalyx are suggested to hold a key functional position in health and disease, considering their potential for binding a great array of proteins, including endothelial superoxide dismutase and xanthine oxidase as well as components of the complement cascade [95,96,97,98].

Heparan sulphate containing proteoglycans of the renal endothelial basement membrane have been shown to bind L-selectin and monocyte chemoattractant protein (MCP)-1 and induce monocyte adhesion in kidney associated IRI [99]. Similarly, augmented MCP-1 binding to the HS proteoglycans pertaining to the basement membranes of the renal peritubular capillaries has been identified in kidney graft biopsies immediately post-transplantation [99]. Ongoing research will reveal whether HS moieties of the endothelial glycocalyx display similar properties. Likewise, deficiency in the kidney allograft of N-deacetylase-N-sulfotransferase-1 (Ndst1), a HS modifying enzyme which catalyzes sulfate conjugation to carbohydrates, has been correlated with reduced acute rejection, most probably through interfering with the interaction of glycosaminoglycans and chemokines [100]. Both augmented and deficient sulphation of heparin moieties in the glycocalyx of kidney grafts have been associated with chronic fibrosis and potentially inflammatory endoglycoside heparanase degradation of the glycocalyx respectively [100,101].

Heparanase is the enzyme which cleaves the glycosidic bond within the HS moieties bound to glycocalyx proteoglycans as well as those pertaining to the extracellular matrix proteins [102]. Heparanase activity is tightly regulated by syndecan-1 and vice-versa, HS and heparanase regulate syndecan-1 shedding [103,104]. Heparanase is considered to play a pivotal pro-inflmmatory and pro-fibrotic role in various disease processes, including AKI and proteinuric kidney diseases, partly as a result of the release of an array of growth factors and cytokines which are normally bound to HS following its degradation [105,106,107]. Thus, heparanase has a direct role in the FGF-2 induced EMT of tubular cells through syndecan-1 mediated fibroblast growth factor (FGF)-2 signaling [106]. In renal transplant recipients, significantly elevated urinary heparanase levels are associated with both proteinuria and graft dysfunction [108]. Increased expression of heparanase not only by the vascular endothelium but by infiltrating CD4+ and CD8+ T cells as well has been linked to acute cellular rejection in murine cardiac allografts. Likewise elevated plasma heparan sulfate levels have been detected in human kidney transplant recipients, before the establishment of renal allograft rejection diagnosis by biopsy, thus supporting the role of heparan sulfate as an early marker of cellular rejection [109].

Immunofluorescence staining of renal tissue from a mouse model in which IRI was induced by bilateral clamping of renal arteries, showed evidence of heparanase upregulation at both glomerular and tubulointerstitial sites 72 h following reperfusion [110]. Furthermore, in transgenic mice over-expressing heparanase but not in wild-type mice, IRI induced a significant up-regulation of markers of EMT such as alpha smooth muscle actin (α-SMA) and vimentin [110]. Treatment with a heparanase inhibitor of both Wild type (WT) and heparanase-silenced renal tubular cells submitted to hypoxia and reoxygenation, did not induce significant changes in syndecan-1 expression. Nevertheless, further research is required so as to establish a certain and direct evidence regarding the link between heparanase upregulation in the setting of IRI following kidney transplantation, its cleavage products and clinical outcomes [110].

Experimental evidence suggests that heparanase holds a key position in the process of the recruitment and activation of macrophages in response to IRI and in specific the M1 macrophage polarization profile [111]. M1 macrophages express proinflammatory cytokines such as IL-1b, IL-6 and TNF-α as well as induce the mechanism of EMT in renal tubular cells. Additionally, heparanase augments the expression of TLRs in tubular epithelial cells, vascular endothelial cells, and infiltrating leukocytes during renal IRI, thus creating a positive pro-inflammatory feed-back which eventually leads to tubular cell apoptosis, immune activation, graft rejection and eventually chronic allograft nephropathy [111]. Inhibition of heparanase both in vivo and in vitro diminishes the M1 macrophage response pathways without affecting the M2 macrophages or the expression of the M2 markers, such as Arginase1 and the macrophage mannose receptor (MR). M2 markers are associated with anti-inflammatory and immune-modulating responses as well as promotion of tissue repair. This would consequently translate into improved histological patterns and renal function as indicated by experimental evidence from mice subjected to IRI [111].

Similarly, IRI induces a long-term over-expression of heparanase by the kidneys, following the initial insult, which is compatible with the establishment of chronic allograft nephropathy in kidney transplantation. Gene expression analysis and immunofluorescence staining of kidney tissue from mice with unilaterally induced renal IRI, revealed amplified expression of heparanase in the glomeruli and in the interstitial cells even 8 weeks after the procedure of unilateral renal artery clamping [112]. This was associated respectively with increased collagen accumulation, up-regulation of MMP-2 and MMP-9, increased TNF-α, IL-1b and IL-6 gene expression as well as higher renal and plasma levels of malondialdehyde, a lipid peroxidation product [112]. On the other hand, administration of Roneparstat, a heparanase inhibitor, abrogated all the above effects.

Experimental data show that IRI in the kidneys is associated with reduced expression of endothelial NOS (eNOS) and simultaneously with increased inducible NOS (iNOS) and endothelin-1 expression by the renal endothelium and the inflammatory cells [113,114,115,116]. There seems to be a close relation between the mediators of endothelial dynamics, such as endothelin-1 and nitric oxide synthases (NOS) with heparanase. Accordingly, eNOS appears to prevent heparanase induction in a model of proteinuric kidney disease whereas heparanase inhibition blunts inducible NOS (iNOS) and endothelin-1 production by the renal endothelium in the setting of IRI [113,114].

As already described previously, hyaluronan is a ubiquitous glycosaminoglycan not only pertaining to the extracellular matrix but to the endothelial glycocalyx as well, albeit accounting for less than 20% of its glycosaminoglycan content. Hyaluronan significantly contributes to the endothelial glycocalyx thickness and structure preservation. It regulates mechanical signal transduction to the endothelial cells through flow-mediated NO production as well as the endothelial permeability to white blood cells and platelets [117,118,119].

Animal models of severe renal IRI to a single kidney, thus simulating the conditions of renal allograft transplantation, indicate sequential biphasic induction of hyaluronan synthases 1 and 2 in the renal tissue, which is manifested by transient increase in high molecular weight hyaluronan deposition followed by a delayed accumulation of lower size hyaluronan products [120]. Low molecular weight hyaluronan fragments appear to be implicated in the inflammatory cascade through activation of toll-like receptor-4 (TLR4) and -2 (TLR2) as well as in the genesis of renal fibrosis [32,120]. Low molecular weight hyaluronan fragments following interaction with the CD44 hyaluronan receptor, cause enhanced actin fiber formation in the endothelial cells and disruption of the endothelial barrier, which is characterized by capillary ballooning, mesangiolysis, and loss of endothelial fenestration [117,121,122].

Inactivation of the hyaluronan synthesizing enzyme, hyaluronan synthase 2 in endothelial cells of mice led to more than 50% loss of the glycocalyx structure compared to control mice, as estimated by cationic ferritin coverage, although the rest glycocalyx constituents were not affected [57].

The interaction of hyaluronan with its receptor CD44 has been implicated in pathophysiology of IRI with stimulation of macrophage recruitment by inducing expression of monocyte chemoattractant protein-1 (MCP-1) by the renal tubular cells as well as through promoting renal fibrosis via the transforming growth factor (TGF)-β pathway [122,123,124]. In rat models of IRI, significant ectopic up-regulation of hyaluronan synthase 2 expression by the renal cortex together with accumulation of cortical hyaluronan up to ten times its normal amount was observed [125].

Although CD44 is barely expressed in the renal tissue under normal conditions, it is markedly and rapidly upregulated in the infiltrating white blood cells as well as the capillary endothelial cells and renal tubular epithelia in postischemic kidneys [126,127,128,129]. Available experimental evidence indicates that adhesion and migration of neutrophils in the setting of IRI, is mediated by the interaction of membrane-bound hyaluronan moieties expressed by the neutrophils with the de novo expressed CD44 on the renal endothelial cells [126].

It should be noted that there is a continuous prominent expression of CD44 by the endothelial cells of renal allografts both under normal conditions as well as with acute rejection, which is otherwise not evident in native kidneys [130]. Deficiency of hyaluronidases, the enzymes responsible for hyaluronan degradation exacerbates renal damage in the post-ischemic kidney [131]. Pharmacological inhibition of hyaluronan synthesis in the setting of IRI is associated with a marked decrease of the content of hyaluronan and of CD44 expression in the renal tissue, as well as of the inflammatory infiltrate in the post-ischemic kidney, which translates improved renal function [132]. Likewise, CD44 absence or its pharmacological inhibition result in decreased influx of neutrophils attenuated kidney injury and preserved renal function following IRI [126].

Hyaluronan moieties in the endothelial glycocalyx also specifically bind to Agiopoetin 1 through a lectin-like fold, a linkage which is a prerequisite for Angiopoetin 1 binding to the glomerular endothelium via its Tie2 receptor [57]. Angiopoetin 1 is an angiogenic factor secreted by a multitude of cells, including endothelial cells, vascular smooth muscle cells and mesenchymal cells, which possesses anti-inflammatory as well as antiapoptotic properties. Following IRI, the renal Angiopoetin1 expression begins to increase after 7 days and is sustained for at least 14 days after IRI, suggesting its role in the neo-angiogenesis of the repair process [133]. Experimental models of renal IRI indicate that Angiopoietin-1 promotes the mobilization and recruitment of endothelial progenitor cells in the kidneys, thus attenuating the effects of IRI [134]. Furthermore, administration of COMP-Ang1, an engineered variant of angiopoietin-1 in mice with renal IRI reduced the infiltration of neutrophils and macrophages in the kidneys, preserved renal tissue perfusion and microvascular permeability as well as decreased interstitial fibrosis [135].

### 5.3. Novel Insights: Sphingosine-1-Phosphate Signaling in IRI and the Endothelial Glycocalyx

Sphingosine 1-phosphate (S1P) is a sphingolipid with a plethora of physiologic roles, mediated mainly by interacting with its five subtypes of G-protein coupled receptors (S1PR1-S1PR5), which are differentially distributed in specific tissues [136]. S1P acts both as an intracellular messenger regulating processes such as cellular proliferation and apoptosis, as well as an autocrine and paracrine agent. The major carrier of S1P in the plasma is the HDL molecule. In the setting of IRI, S1P is released by a variety of cells, including platelets, endothelial cells, and leukocytes where it modulates endothelial permeability and immune cell infiltration through its S1PR signaling pathways [15,136,137]. S1P itself and S1P agonists have been shown to play a protective role in various IRI models, including myocardial, pulmonary and liver IRI [138,139,140]. S1P exerts its pleiotropic nephroprotective effects in kidney IRI, through regulation of endothelial hemodynamics, protection of tubular epithelial cells from apoptosis and above all immune modulation [141,142,143]. It has been shown that expression of the S1PR in renal endothelial cells, peaks 3 h after IRI [144].

In the setting of ischemic AKI, mice with deletion of endothelial S1P1R displayed increased expression of pro-inflammatory mediators such as ICAM-1, MCP-1 and TNF-α, impaired vascular permeability, as well as more severe patterns of renal tubular necrosis and apoptosis compared to mice with normal S1P expression [145,146]. It has been suggested that the protective role that endothelial S1P1R exerts against ischemic AKI is at least partly mediated by regulating heat shock protein (HSP) 27 expression, which is well known for its cytoprotective functions [145,146].

There is substantial evidence supporting a role for S1P protection of the endothelial glycocalyx and subsequently maintenance of endothelial permeability, as well as it boosting glycocalyx recovery following injury [147,148]. In a cell culture model of rat fat-pad endothelial cells, not only was the protective effect of plasma proteins to the structural stability of the endothelial glycocalyx confirmed, but it was also demonstrated that this effect was in fact mediated by plasma protein bound S1P interaction with its S1P_1_ receptor [147]. Accordingly, activation and phosphorylation of the S1P_1_ receptor by S1P inhibits the activity of MMP-9 and MMP-13 possibly via Rac-1-dependent pathways. As a result shedding of syndecan-1 ectodomain as manifested by chondroitin sulphate and heparin sulphate losses is suppressed [147].

It should be noted that even in the absence of S1P carrier proteins, exogenous administration of S1P appears to protect the glycocalyx from shedding [147]. Furthermore, evidence from cell culture studies indicates that S1P induces synthesis of glycocalyx through the phosphatidylinositol-3 kinase–dependent (PI3K) signaling pathway and thus promotes its recovery following injury. The PI3K-Akt signaling axis is induced by several mediators in the endothelial cells, including VEGF and S1P, and it is crucial for regulation of eNOS activity as well as endothelial cell survival and migration [149,150]. In vitro experiments of glycocalyx degradation have shown that exogenous administration of heparin sulphate together with S1P restores both the glycocalyx structure as well as gap junctions among endothelial cells [151].

Addition of S1P to a functional tissue-engineered blood vessel constructed by human endothelial cells and human cord blood derived endothelial progenitor cells on a decellularized human umbilical vein scaffold, resulted in enhanced syndecan 1 expression on the human endothelial cells which was accompanied by attenuated platelet adherence to the endothelium [152]. Similarly, human umbilical vein endothelial cells exposed to shock conditions demonstrated increased shedding of syndecan-1 and hyaluronic acid, which diminished following administration of S1P enriched plasma [153].

Still, the link between S1P signaling and glycocalyx status during IRI relies mainly on experimental data, at times controversial. Thus, recent evidence from a rat model of heart IRI showed that although IRI undoubtedly increased the release of syndecan-1 in the coronary effluent, treatment with S1P before development of ischemia had no visible effect on syndecan-1 release [154]. Still, the authors of the study suggest that the concentration and timing of S1P administration might have affected the aforementioned results.

## 6. Conclusions

The endothelial glycocalyx is a unique microenvironment and its integrity is of pivotal importance for organ function. Ongoing progress in understanding the complex impact of IRI on the endothelial glycocalyx, opens up a new era of research in the field of organ transplantation. Although recent advances in technology are making the visualization of the endothelial glycocalyx and the elaborate analysis of its components possible, currently available evidence relies mostly on experimental data and straightforward conclusions cannot always be drawn. Clinical studies evaluating the diagnostic and prognostic value of markers of endothelial glycocalyx damage either in the peripheral circulation or in kidney allograft biopsies are of utmost importance in the future. Furthermore, future research shall shed light on the intertwining pathophysiological pathways underlying the alterations of the endothelial glycocalyx in the setting of kidney transplantation, which would be crucial for exploring potential therapeutic targets.

## Figures and Tables

**Figure 1 ijms-22-02157-f001:**
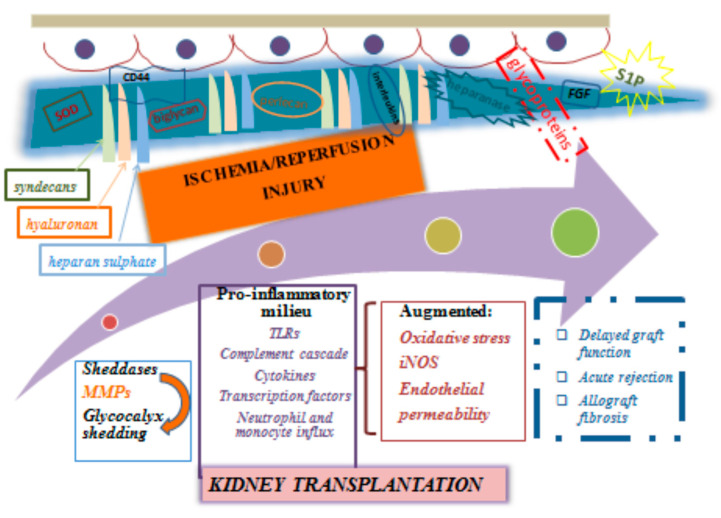
Clinical implications of glycocalyx damage in the setting of ischemia and/or reperfusion injury following kidney transplantation. FGF, fibroblast growth factor; iNOS, inducible nitric oxide synthase; MMPs, matrix metalloproteinases; S1P, sphingosine 1-phosphate; SOD, superoxide dismutase, TLR, toll like receptors.

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
