# Peer review of "The Endothelial Glycocalyx as a Target of Ischemia and Reperfusion Injury in Kidney Transplantation—Where Have We Gone So Far?"

_ijms, 2021, doi:10.3390/ijms22042157_

Round 1

Reviewer 1 Report

The endothelial glycocalyx as a target of ischemia and reperfusion injury in kidney transplantation – where have we gone so far?

General evaluation

This paper addressed a very important topic for transplantation: The damage of the endothelial glycocalyx is potentially associated with delayed graft function, acute rejection as well as long-term allograft dysfunction

The main aim of this review was to present the currently available data regarding complex links between shedding of the glycocalyx components, like syndecan-1, hyaluronan, heparan sulphate, and CD44 with the activation of intricate immune system responses, including toll-like receptors, cytokines and pro-inflammatory transcription factors.

The secondary aim was to depict evidence on modes of protection of the endothelial glycocalyx and subsequently maintenance of endothelial permeability as well as novel nephroprotective molecules such as sphingosine-1 phosphate (S1P).

A review is something else than a systematic review however a method section is needed.

Please check out the PRISMA Guidelines:

http://www.prisma-statement.org/PRISMAStatement/Checklist

The registry is a quality check and a standard. I know that this is a review and not a systematic review however a minimum of methods is needed.

This review is important and has relevance for clinicians and researchers.

Following elements can be considered to strengthened the paper further:

Title

ok

Abstract

The abstract provides content information. There is not enough method information in the abstract.

Introduction

  • Nice introduction
  • Aim is slightly different than in the abstract. Please adapt.
  • The aim was to present the currently available data regarding the consequences of IRI on the renal endothelial glycocalyx of kidney graft, so as to create a frame not only for potential clinical implications but for future therapeutic targets as well.
  • To create a frame a strong review method is needed.

Material and Methods

There is a whole bunch of information missing. Without these information, I can not trust your results. Therefore I ask your research team to add all the information needed:

  • Did you ask a librarian to help to find the right search Term and then the right string? Please add the used search string and the people involved
  • Which databases did you include? What about grey literature?
  • What was the review protocol? How many studies were included? Why were some studies not included?

Results

  • Nice reporting of the studies. Nice image.

Discussion

  • Please add more precisely what would be the next study that should be done. Which design ? Inclusion criteria?
  • Implication for practice: There are much more implication for Practice – Intensive care / Sepsis….

Author Response

We would like to thank the reviewer for his comments.

As the reviewer has commented, the nature of our article is a narrative review and not a “systematic review”. Considering that the until now available published data on the implications of ischemia/reperfusion injury on glycocalyx damage in the field of organ transplantation are mainly of experimental nature and simulations of one-kidney acute ischemic injury models, as presented in the manuscript.

Nevertheless, although we cannot apply “PRISMA” guidelines in our narrative review, based on the suggestions of the reviewer, we added an “Aims and methods” section after the “Introduction” section in which we provide the following information:

-We removed the term “frame”.  

-We adapted the aim as to be similar to the one provided in the “Abstract” as required. “The aim of our review is to present the currently available data regarding complex links between shedding of the glycocalyx components, like syndecan-1, hyaluronan, heparan sulphate, and CD44 with the activation of intricate immune system responses, including toll-like receptors, cytokines and pro-inflammatory transcription factors. Evidence on modes of protection of the endothelial glycocalyx and subsequently maintenance of endothelial permeability as well as novel nephroprotective molecules such as sphingosine-1 phosphate (S1P), are also depicted”.

-Additionally we added the search terms: Accordingly, we searched the electronic databases including PubMed, Medline, and Cochrane for all publications on solid organ transplantation or kidney/renal transplantation, and ischemia and reperfusion injury and acute kidney injury and endothelial glycocalyx, syndecan, hyaluronan, heparan sulphate, CD44, until November 2020. We included both experimental and original clinical studies. Additionally, we hand-searched the references of every relevant study and review article for additional publications”.

Moreover, we state that we did not use references form grey literature.

Although we suggest in the “Conclusions” section that clinical studies should be conducted in the future considering that most evidence currently is mainly experimental.  We believe that this is not within the purpose of our review, to suggest a possible clinical study design on the field. Moreover, this is a vast field and it would be impossible to summarize potential future suggestions for clinical research. Future research will probably focus on glycocalyx markers both in the peripheral circulation and in kidney allograft biopsies of kidney transplant recipients and their role as potential prognostic markers for kidney allograft dysfunction. Moreover, new studies evaluating therapeutic methods for glycocalyx protection are required.

-Accordingly, we modified in the “Conclusions” section the following sentence:Clinical studies evaluating the diagnostic and prognostic value of markers of endothelial glycocalyx damage either in the peripheral circulation or in kidney allograft biopsies are of utmost importance in the future.  Furthermore, future research shall shed light on the intertwining pathophysiological pathways underlying the alterations of the endothelial glycocalyx in the setting of kidney transplantation, which would be crucial for exploring potential therapeutic targets.”

-Finally, regarding implications for other entities as suggested by the reviewer, we present available data from other disease models such as diabetic nephropathy and sepsis and we have cited relevant references such as ref. 96 “Cosimo et al, Glycocalyx and sepsis-induced alterations in vascular permeability. Crit Care. 2015, 19(1), 26”, etc. However, we did not fully depict in detail related data considering that our focus was on kidney transplantation.

Reviewer 2 Report

In this manuscript Anila Duni and Colleagues present a review of the current knowledge regarding the damage of the endothelial glycocalyx in kidney transplantation. In particular, they focus on the role of ischemia and/or reperfusion injury (IRI) as the primum movens of a series of events that characterise the pathophysiology of this setting.
In the first part of the manuscript an overview of the IRI and glycocalyx is succinctly and clearly presented. The core sections of the paper highlight the role of each biological element and the potential clinical implications of their alteration.
Overall the review is focused and reads well thus it is appealing to the wide readership of IJMS   

Author Response

We would like to thank the reviewer for his comments.